# Simulation of Heat Flow in a Synthetic Watershed: Lags and Dampening across Multiple Pathways under a Climate-Forcing Scenario

**Daniel T. Feinstein [1],*, Randall J. Hunt [2] and Eric D. Morway [3]**

[1]   U.S. Geological Survey Upper Midwest Water Science Center, 3209 North Maryland Avenue, Milwaukee, WI 53211, USA

[2]   U.S. Geological Survey Upper Midwest Water Science Center, 1 Gifford Pinchot Drive, Madison, WI 53726, USA

[3]   U.S. Geological Survey Nevada Water Science Center, 2730 N. Deer Run Road, Carson City, NV 89701, USA

*   Correspondence: dtfeinst@usgs.gov

**Abstract:** Although there is widespread agreement that future climates tend toward warming, the response of aquatic ecosystems to that warming is not well understood. This work, a continuation of companion research, explores the role of distinct watershed pathways in lagging and dampening climate-change signals. It subjects a synthetic flow and transport model to a 30-year warming signal based on climate projections, quantifying the heat breakthrough on a monthly time step along connected pathways. The system corresponds to a temperate watershed roughly 27 km on a side and consists of (a) land-surface processes of overland flow, (b) infiltration through an unsaturated zone (UZ) above an unconfined sandy aquifer overlying impermeable bedrock, and (c) groundwater flow along shallow and deep pathlines that converge as discharge to a surface-water network. Numerical simulations show that about 40% of the warming applied to watershed infiltration arrives at the water table and that the UZ stores a large fraction of the upward-trending heat signal. Additionally, once groundwater reaches the surface-water network after traveling through the saturated zone, only about 10% of the original warm-up signal is returned to streams by discharge. However, increases in the simulated streamflow temperatures are of similar magnitude to increases at the water table, due to the addition of heat by storm runoff, which bypasses UZ and groundwater storage and counteracts subsurface dampening. The synthetic modeling method and tentative findings reported here provide a potential workflow for real-world applications of climate-change modeling at the full watershed scale.

**Keywords:** heat transport; watershed modeling; temperature; climate change

## 1. Introduction and Objectives

As the climate warms, researchers are increasingly focused on characterizing the effects of atmospheric change on different parts of the natural environment, including surface and subsurface pathways within a watershed (Figure 1). The warming of a groundwater/surface-water system is conditioned by two primary factors. First, the top of the system is separated from the warming atmosphere by an unsaturated zone (UZ). The UZ transmits and stores water and heat as they move downward to the water table [1,2]. It acts as a low-pass filter on water and heat impulses integrated over time by lagging and dampening the thermal load after it leaves the bottom of the root zone. The influence of this filtering is influenced by the thickness of the UZ [3]. Second, in temperate regions, younger groundwater stored near the top of the saturated zone can have different temperatures than older groundwater from deeper parts of the aquifer. As these flow paths converge near stream, lake, and wetland discharge locations [4], the total amount of heat transmitted back to the surface-water system is determined by the combined effect of all

the water pathways. Such factors influence the timing, magnitude, and distribution of the thermal energy that eventually discharges to the surface-water system through distinct watershed pathways. Transient effects can be difficult to evaluate using qualitative analyses of downstream receptors but are crucially important for realistically forecasting system response to future temperature increases. Such increases—even when modest—can form tipping points that transform surface-water ecology and habitats [5] and change subsurface nutrient cycling [6].

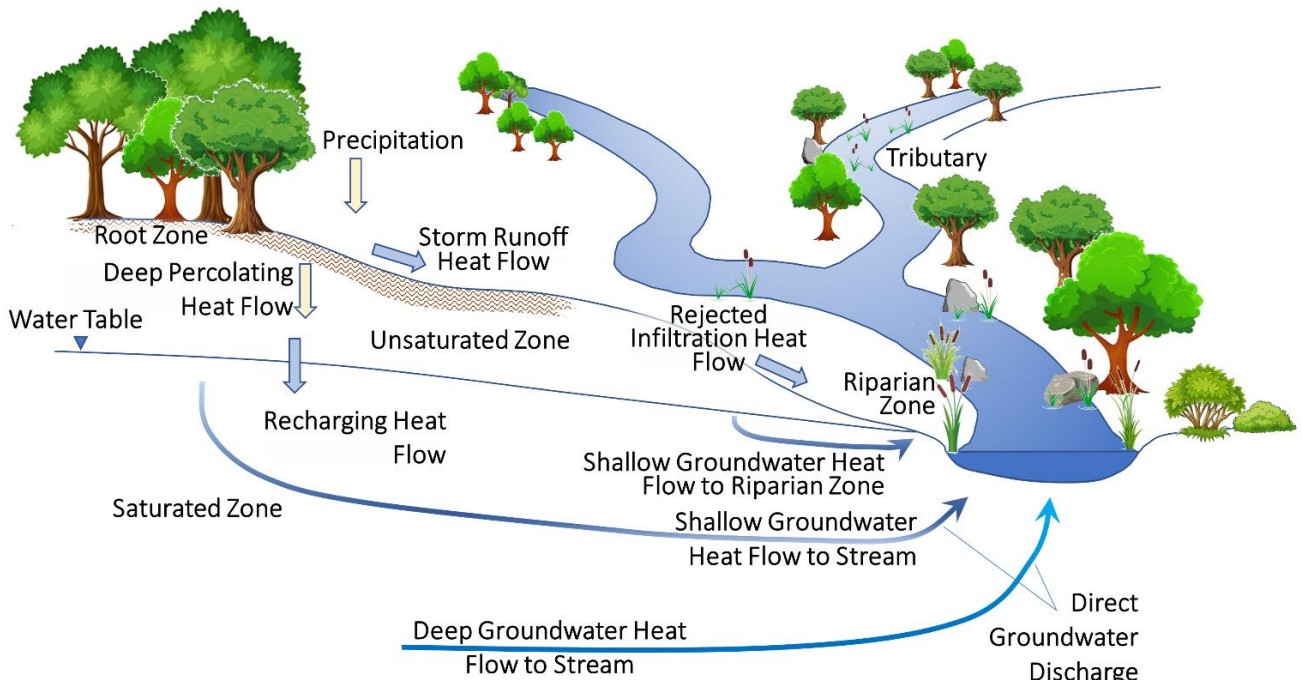

**Figure 1.** Schematic of heat-flow pathways through watershed.

Because downstream aquatic habitats typically integrate influences from upstream, understanding the potential effects of climate change on water resources requires studies and simulations at a watershed scale. The conditions that influence watershed heat transport pathways include the variable thickness of the UZ, variation of groundwater residence time with depth, and topographic/geographic factors such as the width of riparian areas and stream density of the surface-water system. In addition, quantitative studies must also account for different flow and transport processes along these system pathways, for example, the propagation of heat by convection, conduction, and dispersion.

The translation of atmospheric warming to aquatic resources has been a focus of previous work, including analytic [7], process-based watershed [8–10], regression-based [11–13], remote sensing [14], and field measurement [15] approaches. Our work extends this rich history through transient quantitative numerical simulation of salient watershed pathways that store and transport heat. Specifically, this study leverages recent advances in quantitative transient numerical methods detailed in Morway et al. [1] and builds upon initial testing of the synthetic watershed presented in Morway et al. [2]. Here the previous work is extended in three important ways:

1.  Heat transport at monthly intervals is explicitly tracked at the watershed scale (1) between the top of the UZ and the water table, (2) between the water table and groundwater discharge zones, and (3) from upstream to downstream in the surface-water network fed by groundwater. The quantification of the heat-flow across various boundaries within a watershed, for example, the water table, enables a more detailed evaluation of the thermal response of a watershed to a changing climate, and in particular to warming infiltration. This type of analysis will be increasingly important as the thermal impacts of a changing climate affect, for example, cold-water fisheries.

2.  Whereas Morway et al. [2] employed a heuristic synthetic heat inflow time series to isolate distinct warming effects, this study uses a climate forcing function based on the "mean model" Global Climate Model (GCM) high-emission scenarios [16]. The forcing combines the effects of, first, monthly temperature changes and, second, monthly changes in precipitation that are carried into changes in the infiltration rate through the root zone to the top of the UZ. Application of a specific climate scenario permits a more realistic lag [17] and dampening assessment, which in turn facilitates an extension of these methods to real-world decision support settings.

3.  In most watersheds, stream flow is a combination of surface (e.g., overland runoff) and subsurface flows. The transfer of heat to the stream network is therefore dependent on both pathways. Variation and extremes observed in stream temperatures tend to be much greater than what is observed in ambient groundwater temperatures-even in baseflow-dominated systems [18,19]. The difference is attributable in large measure to the influence of quick-flowpath additions of storm runoff during warm, wet months that overprint slower/steadier rates of groundwater thermal discharge. Therefore, this analysis expands upon Morway et al. [2] by simulating and analyzing the heat load returned to the surface-water network from storm runoff in addition to the heat load returned by the groundwater system.

The simulations under study use the groundwater flow model MODFLOW-NWT [20] and a recently augmented version of the companion transport code MT3D-USGS [1,21]. Explicit simulation of heat transport through the UZ makes this work distinct from previous watershed-scale efforts [22–24]. The model output includes the two dependent variables head and temperature, as well as volumetric water and heat fluxes, all of which have utility for watershed-scale assessments.

The methods, results and discussion presented in this article are accompanied by information in a Supplementary Material Section [25]. It consists of three appendices giving additional detail (including Supplementary Figures and Tables) on subjects referenced below.

## 2. Methods

Heat flow travels through the watershed via linked thermal pathways (Figure 1). For example, subsurface heat loading often begins with heat inflows that originate as infiltration below the bottom of the root zone. The infiltrating heat next moves downward through the UZ to the water table and, upon recharging the aquifer, begins migrating toward a discharge location via shallow and deep groundwater flow paths. Heat associated with precipitation that is unable to infiltrate the subsurface (when the water table is at/above the land surface, or the precipitation rate is faster than the soil's ability to infiltrate) flows more quickly to surface-water features. In this effort, we simulate and analyze heat flow pathways in the synthetic model to illustrate the occurrence and magnitude of lags (changes in phase) and dampening (change in amplitude) of atmospheric warming applied at the top of the UZ as it travels through the watershed and associated surface water system.

The pathways shown in Figure 1 correspond to the following flow and storage terms simulated by the model:

*   Groundwater runoff is defined as the sum of groundwater discharge to land surface and rejected infiltration from the land surface under conditions of Hortonian or Dunnian flow.
*   Baseflow is defined as the sum of direct groundwater discharge to surface water plus groundwater runoff.
*   Total streamflow is defined as the sum of baseflow and storm runoff.
*   From a watershed flow system perspective, the following partitioning occurs:
*   Precipitation is partitioned into infiltration, storm runoff and evapotranspiration (ET) from the root zone.
*   Infiltration is partitioned into rejected infiltration, recharge and storage changes in the UZ (no ET is simulated from the UZ since the infiltration is considered to be what percolates below the root zone).

- Recharge is partitioned into storage changes in the groundwater system, direct discharge to surface water and discharge to land surface.

Note that some streamflow generation conceptualizations include an "interflow" subsurface pathway reflecting late-storm seepage that contributes to a recessional limb of a storm hydrograph [26]. Here, however, we focus on more time-integrated results from the watershed (monthly to multi-decadal) and consider such interflow contributions as included in the other subsurface components to streamflow. Moreover, although rejected infiltration is a form of runoff, for purposes of this analysis it is included in the baseflow term, leaving stormflow as the only remaining surface runoff component of total streamflow (because direct precipitation on the stream surface is not simulated).

Figure 1 illustrates this framework by showing the pathways from precipitation and infiltration through the various forms of runoff, recharge and discharge for both water and heat.

*2.1. Construction of Spin-Up and Climate Change Forcing Function*

The amount of heat that enters the top of the UZ is the product of the infiltration rate and its temperature. A brief description of how each time series was generated is offered below. A more detailed description is provided in the Supplementary Material Section S1.

Because the effects of a warming climate cannot be represented by steady state conditions, careful selection of the time discretization and initial conditions used within a transient model are important. Because the time-integrated effects of warming infiltration associated with climate change over a 30-year period of analysis was the focus of this study, monthly timesteps were deemed sufficient to represent the seasonal, random, and non-stationary aspects of the warming infiltration on temperatures in the subsurface. To establish initial condition by the start of the 30-year warming period, a 30 year spin-up period with annually-cyclic infiltration rates and temperatures (i.e., the same values were specified for all 30 Januarys, for example) was employed to ensure a dynamic equilibrium by the start of the warming period (see page 313 in Anderson et al. [27]). After spin-up, the infiltration rates and temperatures continue to vary monthly with a seasonal periodicity, but also have an underlying warming trend and a random noise component. As described in detail in the Supplementary Material Section S1, initial conditions used here differ from those used in the companion study [2]. In this work, both the monthly infiltration and its specified temperature time series varied during spin-up. In the companion paper, temperature varied monthly while the infiltration rate was held constant during its spin-up period at 0.2 m/yr. For this work, the average monthly infiltration rates and temperatures used during spin-up are based on a watershed located in southern Wisconsin, USA [19].

After spin-up, infiltration rates and its accompanying temperature are based on the high-emissions RCP-8.5 climate scenario [16] results for the Midwest United States, which generally reflect wetter and warmer conditions compared to the spin-up period. Downscaled regional results from southern Wisconsin, USA (Figure 2) were applied to the synthetic watershed, where simulated warming corresponds to the period 2022 through 2051. The warming trend applied to the infiltration temperature is imposed on the seasonal signal, which also includes random noise generated from a uniform distribution centered on 0 °C and a range of 4 °C. By the end of the 30-year warming period, the average annual temperature of the infiltration is approximately 2 °C warmer relative to the end of the spin-up period. The amount of heat inflow at the end of the warming period, which is the product of the infiltration rate and its temperature, is roughly 25% higher than the amount of heat inflow at the end of the spin-up period. For our simulations, this value effectively represents an upper limit of the expected climate change in terms of the heat added to the subsurface attributable to wetter and warmer conditions. Most of the increase in heat inflow is due to the trend applied to the infiltration temperature; only a small part is due to the trend applied to the infiltration rate (Supplementary Material Section S1). Because flow and heat transport are simulated separately, spin-up infiltration rates are specified in the

flow model via the UZF1 package [28] while the temperatures assigned to the infiltration are specified in the UZT package of MT3D-USGS [21].

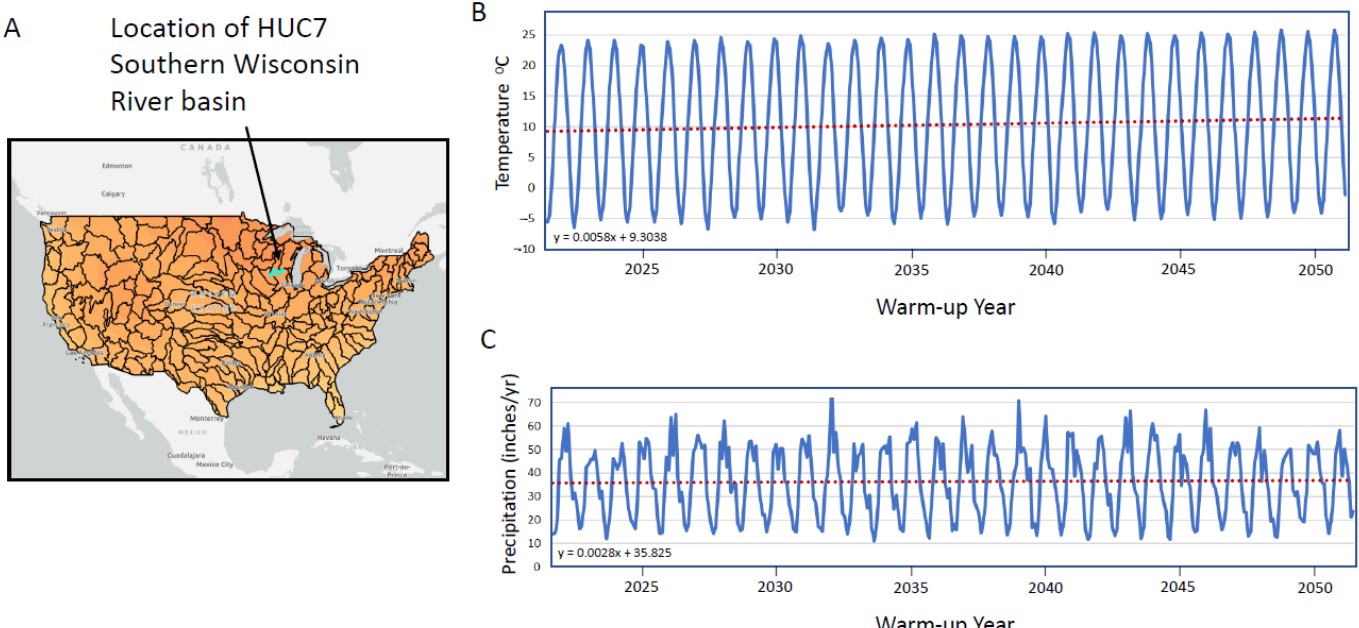

**Figure 2.** Climate forcing used in the synthetic model from high-emissions RCP8.5 scenario [16] downscaled to southern Wisconsin. (**A**) Location of southern Wisconsin basin [29] to which [16] data correspond, (**B**) the atmospheric temperature used to set the temperature of the infiltration (°C), and (**C**) precipitation and infiltration rate forcing (inch/year) where infiltration through the root zone is assumed to be one quarter the precipitation. Equations on plots correspond to dashed linear trend lines. RCP stands for Representative Concentration Pathway.

*2.2. Model Construction*

The synthetic model uses the same geometry, parameter values, and boundary conditions as described in Morway et al. [2]. Additional detail of the model construction is provided in Supplementary Material Section S2. In brief, the salient aspects of the model design are characterized by:

1.  spin-up specification of temporally varying infiltration rates and temperatures that, when multiplied, result in a single time-dependent heat infiltration rate that represents a warming climate signal;
2.  the climate forcing described in (1) is applied in a spatially uniform manner to the entire model domain; that is, infiltration rates and temperatures are temporally variable but spatially uniform;
3.  aquifer/flow and transport parameters [e.g., the hydraulic conductivity (flow) and porosity (transport)], are spatially uniform across the model domain.

The model approximates a mid-sized watershed (about 290 square miles or 750 square km, falling into the HUC10-size category according to the U.S. watershed scheme) that includes streams, wetlands, and a lake (Figure 3). The surface-water system is strongly gaining ("baseflow dominated")-there is very little loss from streams to the aquifer. The subsurface system consists of a sandy aquifer separated from the land surface by an UZ and overlying effectively impermeable bedrock. No-flow boundaries are specified along the east, west, and bottom of the model. Regional groundwater flow gradients from north to south are generated by general head boundaries along the northern and southern model boundaries, but the flow system is strongly influenced by local groundwater divides which reflect the effects of topography and the surface-water network. Cells in layer 1 are typically unsaturated but may contain the water table in riparian zones adjacent to surface-water features. Cells in layers 2 through 4 can

be either unsaturated, partially, or fully saturated. Layers 5 through 8 are fully saturated for the duration of the simulation. Parameter values for the flow and transport simulations are listed in the Supplementary Material Section S1. Monthly stress periods are used in both the flow and transport simulations.

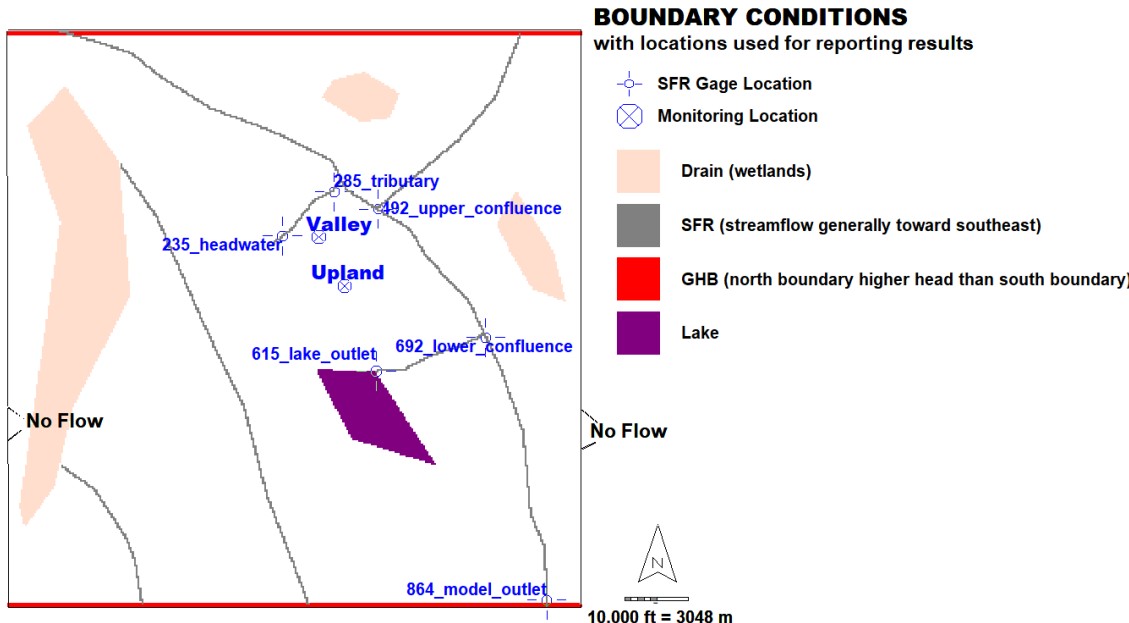

**Figure 3.** Synthetic model setup showing domain and boundary conditions, as well as locations for monitoring temperature results. Cross-sections are shown in Figure 4. [SFR: Stream Flow Routing; GHB: General Head Boundary].

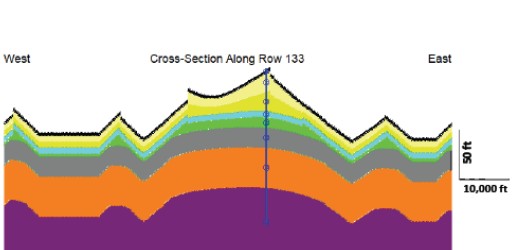

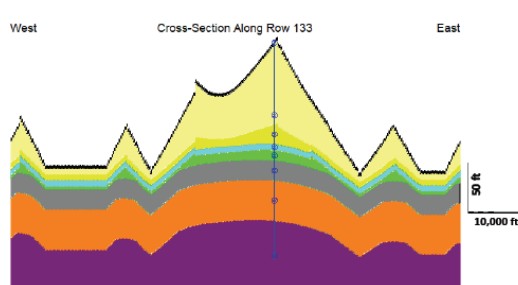

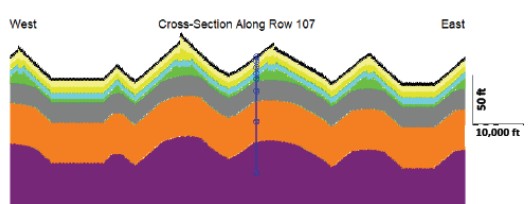

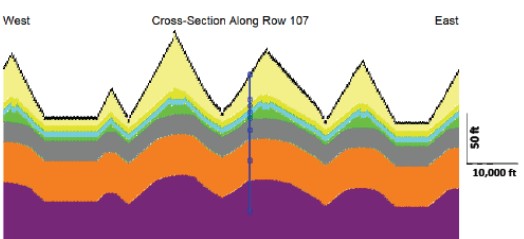

**Figure 4.** Layering through Upland and Valley locations; (**a**) MID_TRENDED model version, Upland cross section. (**b**) MID_TRENDED model version, Valley cross section. (**c**) HI_TRENDED model version, Upland cross section. (**d**) HI_TRENDED model version, Valley cross section. Cross section locations are shown in Figure 3. Top black layer is 3-ft thick receptor layer for receiving infiltration.

An important simplification incorporated in the proposed methodology is to equate the heat signal with infiltration that has *already* passed the root zone. Root zone processes in humid areas, which bear on both the movement of water and heat, include evaporation, transpiration, and conduction, leading to flow that can be both downward and upward. The key assumption in our methodology is that at a monthly transport time step, these root zone processes can be neglected and the warming signal at the top of the UZ can be equated with the average amount of water passing the root zone over the month and with the average monthly atmospheric temperature. This assumption is discussed in detail in Morway et al. [1].

To elucidate the importance of including the UZ in regional-scale heat transport simulations, two versions of the model were constructed for highlighting the effects of UZ thickness on heat transport:

1.  MID_TRENDED model (Figure 4a,b): This version is designed to produce, on average, a moderate water table depth (i.e., moderate UZ thickness) that varies from 0 m in riparian areas (approximately 20% of the model domain) to about 15 m below land surface in the upland areas.
2.  HI_TRENDED model (Figure 4c,d): using steeper topography, this version simulates a thicker UZ compared to the MID_TRENDED model. The water table depth varies from 0 m in riparian areas (approximately 4% of the model domain) to more than 30 m in the upland areas.

Thus, the main difference between the MID_ and HI_TRENDED models is the thickness of layers 1 through 4; the deeper groundwater system represented by layers 5 through 8 is the same in both versions. More information on model construction and model versions, including specification of model flow and transport parameters and selection of parameter values, is provided in Morway et al. [2] and Supplementary Material Section S2.

One of the key differences in the model setup in this analysis compared to that documented in Morway et al. [2] is that surface-water runoff is here explicitly simulated using options available in the UZF1 and SFR [30] packages (Supplementary Material Section S2). Because overland runoff is passed to MT3D-USGS via the linker file [31], it automatically accounts for the heat transported to streams and associated with runoff. Precipitation is assumed partitioned into storm runoff, infiltration, and evapotranspiration. In the synthetic model, storm runoff was set equal to 8.3% of the specified monthly precipitation rate, which is equivalent to 33% of the infiltration rate since the infiltration rate is set to 25% of the precipitation rate. The remaining precipitation is taken up by evapotranspiration rate, equal to 67% of the precipitation. These ratios are intended to represent a porous/sandy watershed where infiltration through the root zone is several times greater than storm runoff. Figure 5 shows the flow budget fluxes for the most upgradient eastern stream subbasin. Whereas the companion analysis [2] focused on understanding the impacts of warming infiltration on baseflow temperatures, this study considers the effects of all return flows on heat transport in the surface-water network, including runoff from the land surface associated with stormflow. The model does not, however, simulate precipitation or evaporation directly on or from the surface water, respectively.

The 75% of the total water and heat flux that enters the watershed over any year (net of evapotranspiration) as infiltration is the source of recharge to the water table. The recharge flux is divided among the following down-basin pathways: groundwater discharge to the stream channels and water bodies, groundwater discharge to land surface (that is, to riparian areas bordering surface water), and rejected infiltration from riparian areas. The portion of these three down-basin terms that terminate as water and heat flux to streams collectively sum to stream baseflow. Stormflow runoff contributes the remaining 25% of the water and heat that enters the watershed (net of evapotranspiration). It runs off instantaneously to the surface-water feature that is directly downslope in the form of a stream segment or lake (Supplementary Material Section S1). The infiltrating heat across the model domain is therefore subject to the low-pass filtering effects (phase and amplitude shifts) of heat transport through unsaturated and saturated flow pathways. Heat

contributed to streams from stormflow runoff is not filtered by the subsurface pathways and is therefore left un-modified (i.e., no lags or dampening is associated with this particular heat transport pathway).

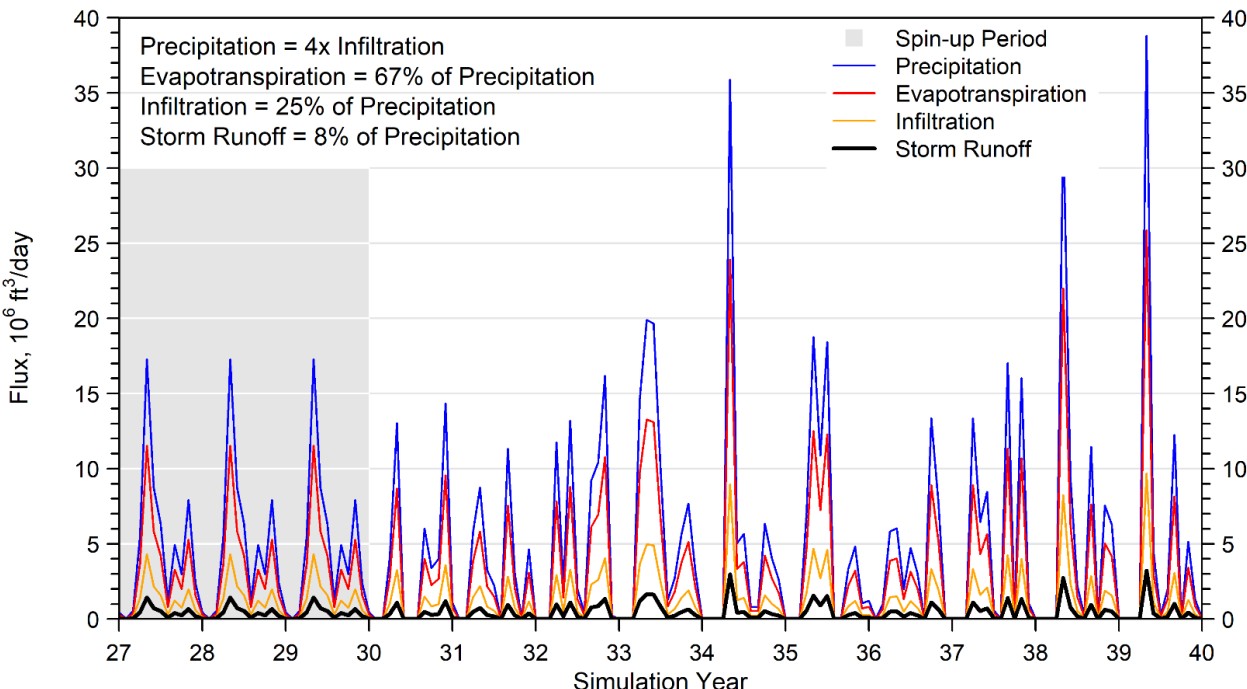

**Figure 5.** Flux terms for an example basin showing assumed relationships of precipitation, evapotranspiration, and storm runoff with infiltration rate at the end of the spin-up and warming periods. The absolute fluxes correspond to amounts for basin contributing runoff to an example stream segment (see Supplementary Material Figure S2-3a for the location of the basin comprising Segment 1 in the synthetic model). Warming begins in Simulation Year 30.

## 3. Results and Discussion

The process-based modeling approach used here produces time series of simulated temperatures throughout the model domain in response to climate forcing, including above the water table (i.e., the UZ), at the water table, in the deep groundwater system, and at various locations in the surface water network. Here, for both versions of the synthetic model, we focus on: (1) the temperature trends along the pathways shown in Figure 1, (2) the distribution, magnitude, and timing of heat transfers (fluxes and flows) within subbasins of the watershed, and (3) the lag and dampening effect of the UZ on the infiltrating heat signal at particular locations and across subbasins within the watershed as well as the lag and dampening effects induced by down-system pathways.

### 3.1. Temperature Trends along Pathways

Although the annual average temperature during the spin-up period is 8.55 °C, the flow-weighted (or infiltration-weighted) average temperature during spin-up is 9.97 °C. At the end of the spin-up period, the simulated stream and lake temperatures converge to about 10 °C for both the simulation with thinner and with thicker UZ thickness (the MID_ and HI_TRENDED simulations, respectively). After reaching dynamic equilibrium conditions by the end of the spin-up period, temperatures assigned to both the infiltration and storm runoff followed the same time-series scheme described above (see section titled "Construction of spin-up and climate forcing function") in the warming period. The average infiltration-weighted temperature over the 30-year warming period is 10.95 °C. Our analysis focuses on the final 10 years of warming, where the infiltration-weighted average temperature was 12.23 °C, a 2.26 °C rise compared to the last year of the spin-up

period. If no lag or dampening occurred, the temperatures throughout the watershed would reflect this higher 12.23 °C temperature.

Warming in the subsurface was observed in the water table at two locations hereafter referred to as the Upland and Valley locations (Figure 3). The thinner UZ associated with the MID_TRENDED model contributes to a greater thermal response at the water table by the end of the simulation compared to the HI_TRENDED result (Figure 6A,B). In addition, a thinner UZ contributes to a flashier thermal response at the water table where the UZ thickness is even smaller (<15 ft) at the Valley location (Figure 6A). At the Upland location, the water-table temperature is smooth and muted in both models with a subtle temperature increase simulated at the water table for the first 22 years of the warming period. During the final 8 years of the simulation period, the temperature response at the water table to the overall warming trend is better defined with a clear rise in water table temperatures in year 52 of the simulation. Before that, limited sensitivity to year-by-year fluctuations in the temperature of the infiltration is shown in the MID_TRENDED simulation and no sensitivity is evident in the HI_TRENDED simulation results (Figure 6B). At the Valley location, where the water-table depth is approximately 3.3 m (11 ft), simulated temperatures in the MID_TRENDED simulation exhibit a much more responsive behavior in the last 10 years of the simulation compared to the HI_TRENDED simulation where the water-table depth averages 9.6 m (31 ft).

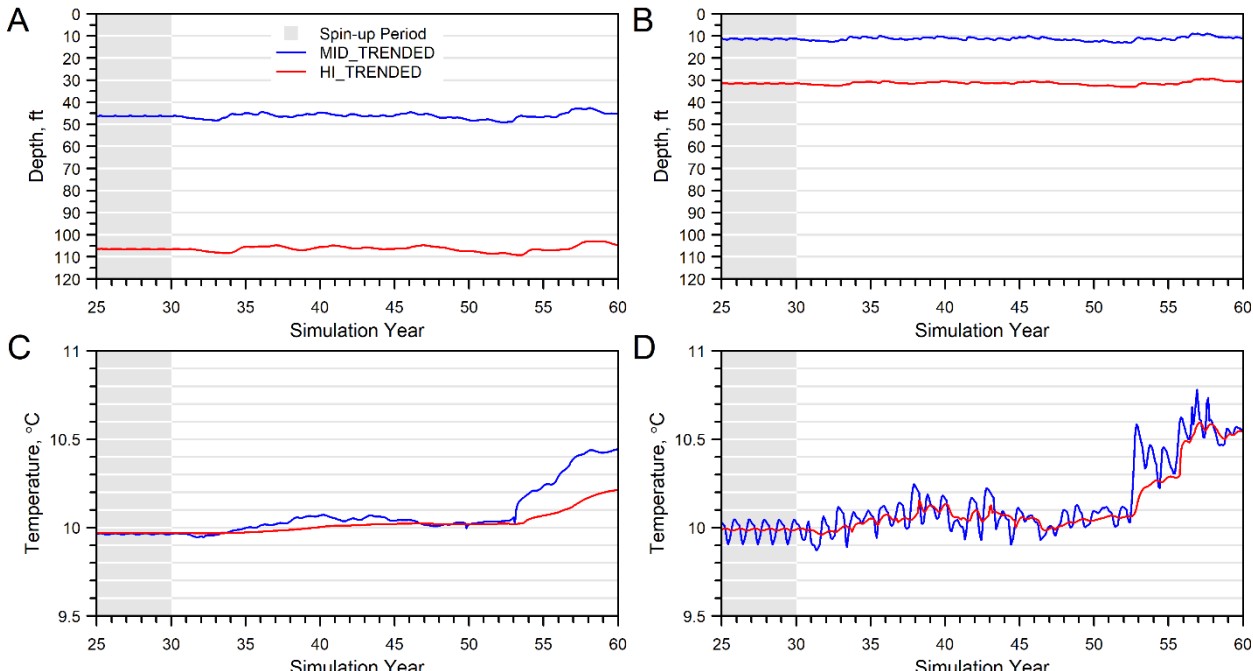

**Figure 6.** Time-series results for Synthetic Model at Upland and Valley Locations: (**A,B**) Depth to water table for MID_TRENDED and HI_TRENDED simulations; (**C,D**) Water table temperature for MID_TRENDED and HI_TRENDED simulations. Upland and Valley locations are shown in Figure 3. Warming period begins in Simulation Year 30.

Figure 7 shows the percent of the model domain with a water-table temperature at a given threshold (*y*-axis) over the warming period (*x*-axis). Red indicates the temperature corresponding to the warmest 20% of the domain, blue correspond to the coolest 20% of the domain. For example, at the beginning of the warming period in the MID_TRENDED simulation (year 0 on the *x*-axis; Figure 7A), the water table temperature across entire model domain is roughly 10 °C, but by the end of the warming period the water table temperature is at or below 10.5 °C. Note also that the MID_TRENDED simulation (Figure 7A) because of its thinner UZ consistently shows flashier water-table temperature responses for the warmest (reds) and coolest (blues) parts of the watershed compared to the HI_TRENDED

simulation (Figure 7B). Direct comparison of the median watershed temperatures through time [represented by the dotted (MID_TRENDED) and solid (HI_TRENDED) contour lines in Figure 7B] indicates that a thicker UZ yields, on average, a more subdued water-table temperature response to warming infiltration, highlighting the ability of a thicker UZ to store and filter heat transport to the water table. As a reminder, the annual average temperature of the infiltration warmed by 2 °C during the 30-year warming period. In response, the shallow groundwater temperatures in the MID_ and HI_TRENDED simulations rose by more than 1 °C across about 20% of the model domain.

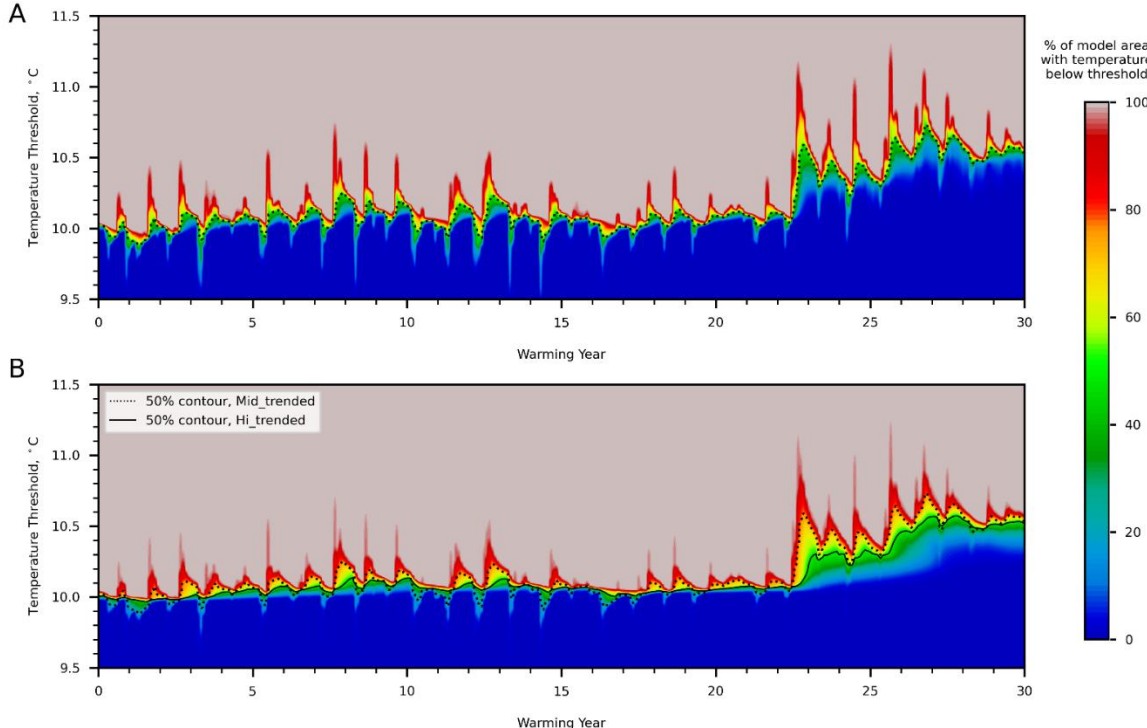

**Figure 7.** The percent water-table area of model domains that is simulated below increasing temperature thresholds (*y*-axis) over the time of the warming period (*x*-axis), for the (**A**) MID_TRENDED and (**B**) HI_TRENDED simulations. The contours indicate the temperature for the 50% threshold. The MID_TRENDED 50% contour displayed in (**A**) also is shown in (**B**) for comparison with the HI_TRENDED 50% contour. Surface-water cells are excluded from calculations.

### 3.2. Heat Fluxes and Heat Flows

It is often instructive to evaluate the response to warming in terms of heat movement instead of temperature change. Heat movement consists of three flux components–convection, conduction and dispersion [1]. A component of heat flux (measured in Watts) normalized by an area perpendicular to the flux direction yields the corresponding component of heat flow (for example, in units of Watts/m$^2$.) There are three main interfaces at the beginning or end of watershed pathways where flux or flow components can be calculated: across the top of the UZ (infiltration), across the water table (recharge) and across a streambed or lake bed (baseflow). They are discussed in turn:

-   In this study the thermal infiltration is equated with the heat movement downward from the bottom of the root zone which occurs after runoff and evapotranspiration have rerouted some of the water and heat along the land surface or to the atmosphere. This net infiltrating heat flow is imposed as a purely downward convective process into the top of the UZ. For our purposes, the combined effect of heat conduction and dispersion at the root zone/UZ interface, either upward or downward, is considered to be unimportant in comparison to the surface and root zone processes that determine the average monthly rate of infiltrating heat flux.

- Our analysis of the relative weight of simulated heat transport components out of the UZ show that for the humid temperate conditions of the synthetic model, the convective heat flow dominates the conductive and dispersive flow at the water-table interface. This relation persists over time at the scale of individual model locations (Supplementary Material Section S3 Figure S3-10) and when averaged over the entire model domain (Figure 8). For the MID_TRENDED simulation, the absolute value of the conductive heat flows at the water table average about 11% of the convective heat flow, whereas the dispersive heat flow is only 0.05% of the convective heat flow. For the HI_TRENDED simulation, incorporating a generally thicker UZ, the corresponding ratios are 7% and 0.03%. It is worth noting that thermal dispersion is a negligible heat flow component owing to the relatively small longitudinal dispersivity specified (0.9 m) relative to the lateral grid spacing [91 m (300 ft)], a choice consistent with a homogeneous synthetic aquifer. These findings suggest that there is in general only minor loss of accuracy if the heat flow across the interface at the top of the groundwater system is approximated by considering the convective heat flux alone.

- The temperature gradient across the streambed between the stream water in the channel and the ambient groundwater could be incorporated in equations that yield convective and dispersive components of heat flow. Thermal conduction would occur whether the temperature gradient is in the same direction as baseflow or in the opposite direction away from the stream; dispersion would occur only when the gradient is in the same direction as the flow through the streambed. However, the MT3D-USGS code neglects these theoretical components of heat flux and only calculates the convective component, either as a function of groundwater temperature in the presence of baseflow or as a function streamflow temperature in the presence of stream.

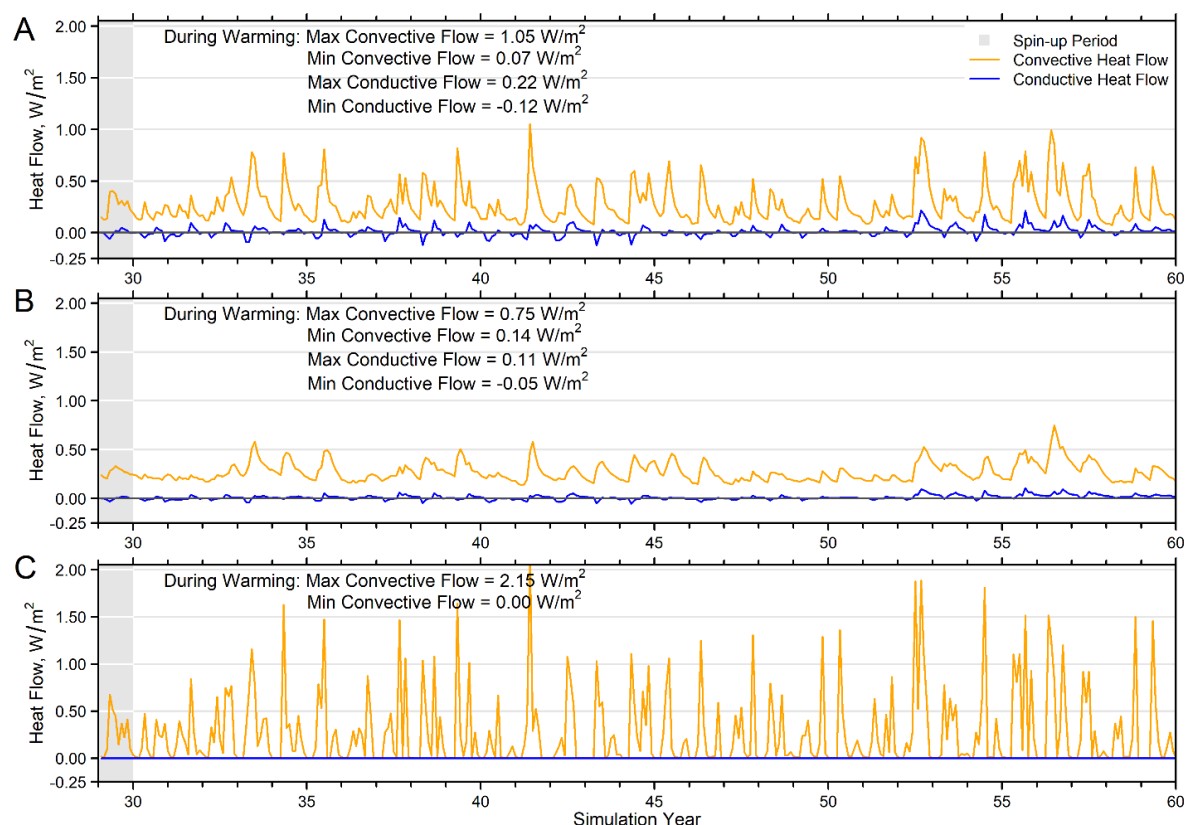

**Figure 8.** Recharge heat flow components (W/m$^2$) averaged monthly over the model domain for the (**A**) MID_TRENDED, (**B**) HI_TRENDED, and (**C**) MID_TRENDED (riparian area only) simulations. Heat flow components are shown for conduction and convection. Dispersive heat flow is negligible. Warming period begins in Simulation Year 30.

Given that heat flux across the major watershed interfaces is either imposed as convective flux, approximated by convective flux, or only calculated for convective flux, it is convenient in the pathway analysis of thermal lagging and dampening presented below to define heat movement strictly in relation to the magnitude and direction of water flow, neglecting the conductive and dispersive fluxes.

The convective flux of heat for any part of the model domain is calculated as the flux of water through the model cells constituting the given volume multiplied by the temperature of the water and by the density and heat capacity of fresh water. A convenient unit for the convective heat flux accumulated over one second is Watts (W, equivalent to 1 joule/sec). The rate of convective heat flow is the flux normalized by the area corresponding to the flow. For example, the heat flow in recharge, baseflow and runoff associated with the upstream areas of gages shown in Figure 3 is equal to the accumulated upstream heat flux divided by the areas reported in Table 1 (also see Supplementary Material Section S2, Figure S2-3b for map view of areas upstream of gages). A convenient unit for the rate of heat flow is Watts per square m ($W/m^2$). Interested readers are directed to Supplementary Material Section S3 for a more detailed discussion of the calculation of the quantities heat flux and heat flow.

**Table 1.** Topographic areas upstream of stream gages identified in Figure 3.

| Stream Gage Number | Gage Description | Upstream Topographic Area | |
|---|---|---|---|
| | | (equated with Gage Recharge, Baseflow and Runoff Areas) | |
| | | mile$^2$ | km$^2$ |
| 235 | Headwater | 2.07 | 5.35 |
| 285 | Tributary | 12.06 | 31.23 |
| 492 | Upper Confluence | 58.62 | 151.82 |
| 615 | Lake Outlet | 30.34 | 78.59 |
| 692 | Lower Confluence | 107.22 | 277.69 |
| 864 | Model Outlet | 134.38 | 348.04 |

Notes: Total area of model domain is 290.5 mile$^2$ = 752.5 km$^2$, taken to be extent of watershed. Upstream area associated with Gage 864 includes entire eastern basin of watershed including all upstream gages.

Convective heat flux diminishes in strength as it moves through the subsurface. The first reduction occurs in the UZ from where heat enters the simulation as infiltration to where it recharges the groundwater system (Figure 9). This loss of heat is largely due to changes in the amount of heat stored in the UZ. Additional losses to the total heat flux through the watershed occur in the saturated zone or as recharge makes its way to discharge locations, for example, as groundwater discharges directly to streams (Figure 9). In this case, as the shallow groundwater is warmed by the recharge associated with warmer infiltration, it mixes with cooler (and deeper) groundwater as it travels through the saturated zone. The effect of mixing is evident in the heat flux results along pathways. In Figure 9, Gage 492 represents integrated conditions over the upper basin of the eastern part of the stream network, and Gage 864 represents conditions for the entire eastern stream network (see Figure 3 for locations). The simulated heat flux in the upgradient stream network (above Gauge 492) is only a fraction of the flux integrated over the entire eastern basin (above Gage 864). However, it is striking that for both gage locations (in both the MID_ and HI_TRENDED versions of the model), the convective heat fluxes entering the subsurface as infiltration, subsequently converted to recharge, increases appreciably over the 30 years due to climate warming. By contrast, the simulated convective heat flux out of the subsurface (i.e., baseflow) increases by a comparatively small amount in response to that warming, pointing to the substantial dampening (from mixing as well as heat storage) that occurs in the saturated zone (Figure 9).

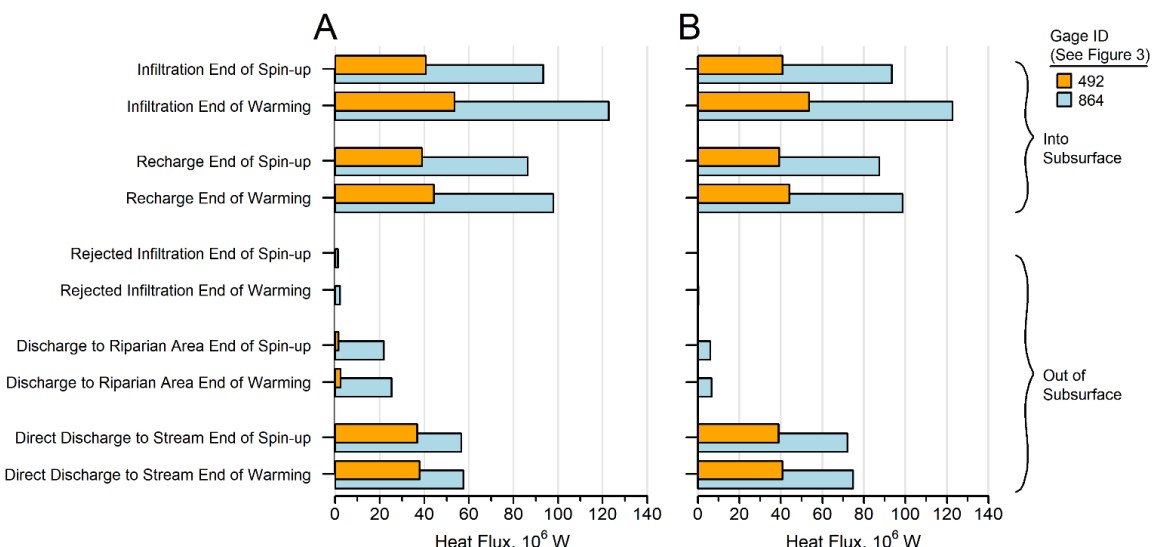

**Figure 9.** Convective heat flux (Watts) accumulated over contributing basins upgradient of two gage locations. (**A**) MID_TRENDED simulation. (**B**) HI_TRENDED simulation. Infiltration flux is compared to flux transmitted by down-system pathways. Convective flux for each pathway corresponds to the average for last year of spin-up ("end of spin-up") and to the average of last 10 years of warming ("end of warming").

Visualization and comparison of results are facilitated by extending the analysis of how simulated heat is propagated across pathways in terms of heat fluxes normalized by the area of model cells or by the area of watershed subbasins. In what follows recharge and discharge thermal transfers are analyzed in terms of heat flows. Heat transmission losses in the UZ (that is, from infiltration to recharge) are primarily the result of heat storage effects due to warming of water in the UZ. A secondary loss of heat can occur when cooler water enters the UZ behind warmer water, producing an upward thermal gradient which gives rise to upward thermal conduction from the deeper part of the system Recall that thermal gradients drive conductive and dispersive flows and can be upward and downward in the UZ whereas the convective flows, given the kinematic wave formulation in the UZF1 packages, only simulates downward flow [1,27,32].

Where the UZ is thin, for example, in riparian areas adjacent to the surface-water features, the infiltrating heat flow readily warms the water table since there is little opportunity to store additional heat in the UZ (Figure 10). During a cool month with moderate infiltration (e.g., March at 15.25 years), the heat flow to the water table is modest, i.e., less than 0.5 W/m$^2$ in both the MID_ and HI_TRENDED models (Figure 10A,B). This is not the case for a relatively warm and wet month (e.g., August at 25.67 years) when the heat flow generally exceeds 2.0 W/m$^2$ in the riparian areas adjacent the surface-water features (Figure 10C,D), though the riparian area is much narrower in the HI_TRENDED simulation. Thus, the spatial distribution of heat flow in the watershed is influenced by the watershed topography, which affects the UZ thickness, as well as by the lateral extent of the riparian area.

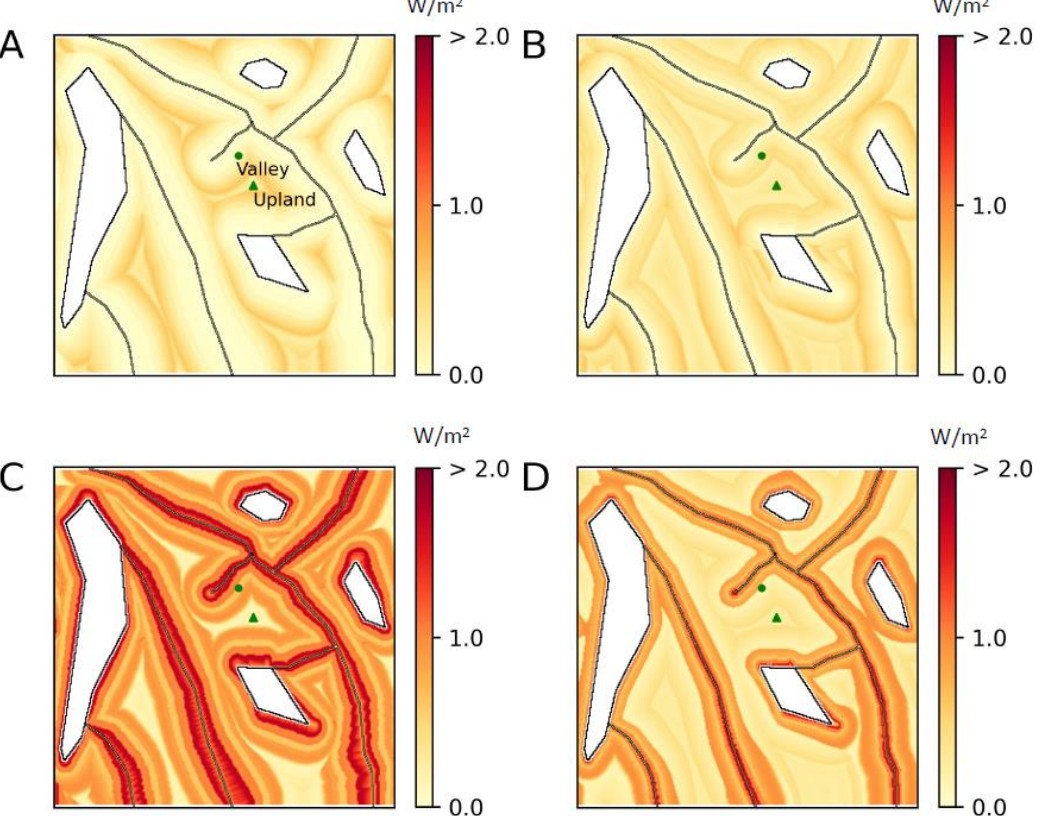

**Figure 10.** Maps of total recharging heat flow at the water table in watts per square meter (W/m²) for the (**A**) MID_TRENDED simulation in the month of March after 15.25 years of warming, (**B**) HI_TRENDED simulation (also in March) after 15.25 years of warming, (**C**) MID_TRENDED simulation in the month of August after 25.67 years of warming, and (**D**) HI_TRENDED simulation (also in August) after 25.67 years of warming. The plotted total heat flow is the sum of the convective, conductive and dispersive heat flows.

*3.3. Lags and Dampening of Heat Signal*

To understand better the role different parts of the hydrologic system have on lagging convective heat transport in the subsurface (that is, changing the phase of the thermal impulse), a lag analysis was performed in terms of correlation coefficients computed at different monthly offsets. The time series of the (causal) infiltrating heat flow was paired with the simulated convective heat flow time series at different locations within the watershed, for example, at the water table, using a set of monthly lags (1, 2, 3, etc. monthly offsets). Correlation coefficients were calculated for each monthly lag and compared across months to yield a measure of the delay in heat transport through different parts of the subsurface system. A separate dampening analysis (that is, the change of amplitude along pathways with respect to the infiltrating signal) was performed by computing the ratio of the average convective heat flow (or temperature in the case of baseflow and streamflow) for the last 10 years of warming to the average convective heat flow (or temperature) value during the last year of the spin-up period. The lag and dampening ratios were calculated for the major pathways shown in Figure 1. Additional details on the calculation procedures are offered in in Supplementary Material Section S3.

The lags (phase) and dampening (amplitude) applied to the infiltrating heat flow prior to its recharging the aquifer is strongly influenced by the thickness of the UZ. The phase and amplitude shifts associated with distinct UZ thicknesses are evident in Figure 11 when comparing the convective heat flow arriving at the water table (red and blue lines) to the heat inflow at the top of the UZ (light blue bars). The heat flow associated with the infiltration at the top of UZ is identical for both runs. At the Upland location, for example,

where a relatively thick UZ exists (Figure 11A), lags are longer with more significant muting compared to the Valley location where the UZ is relatively thin (Figure 11B). The effect of the UZ is further highlighted by contrasting the recharging heat flow only at the Upland location for the MID_ and HI_TRENDED simulations (Figure 11A). That is, the additional UZ thickness in the HI_TRENDED simulation adds months to the arrival time of the infiltrating heat flow at the water table and further subdues the magnitude of the heat flow (Figure 11A, note the peaks of the red line are lower than the peaks of the blue line). The dashed lines in Figure 11, corresponding to the temperature of the water table cell, reflect the effects of the recharging heat flow mixing with water table. At the Upland location, the simulated temperature at the water table is not as responsive (Figure 11A) as at the Valley location (Figure 11B).

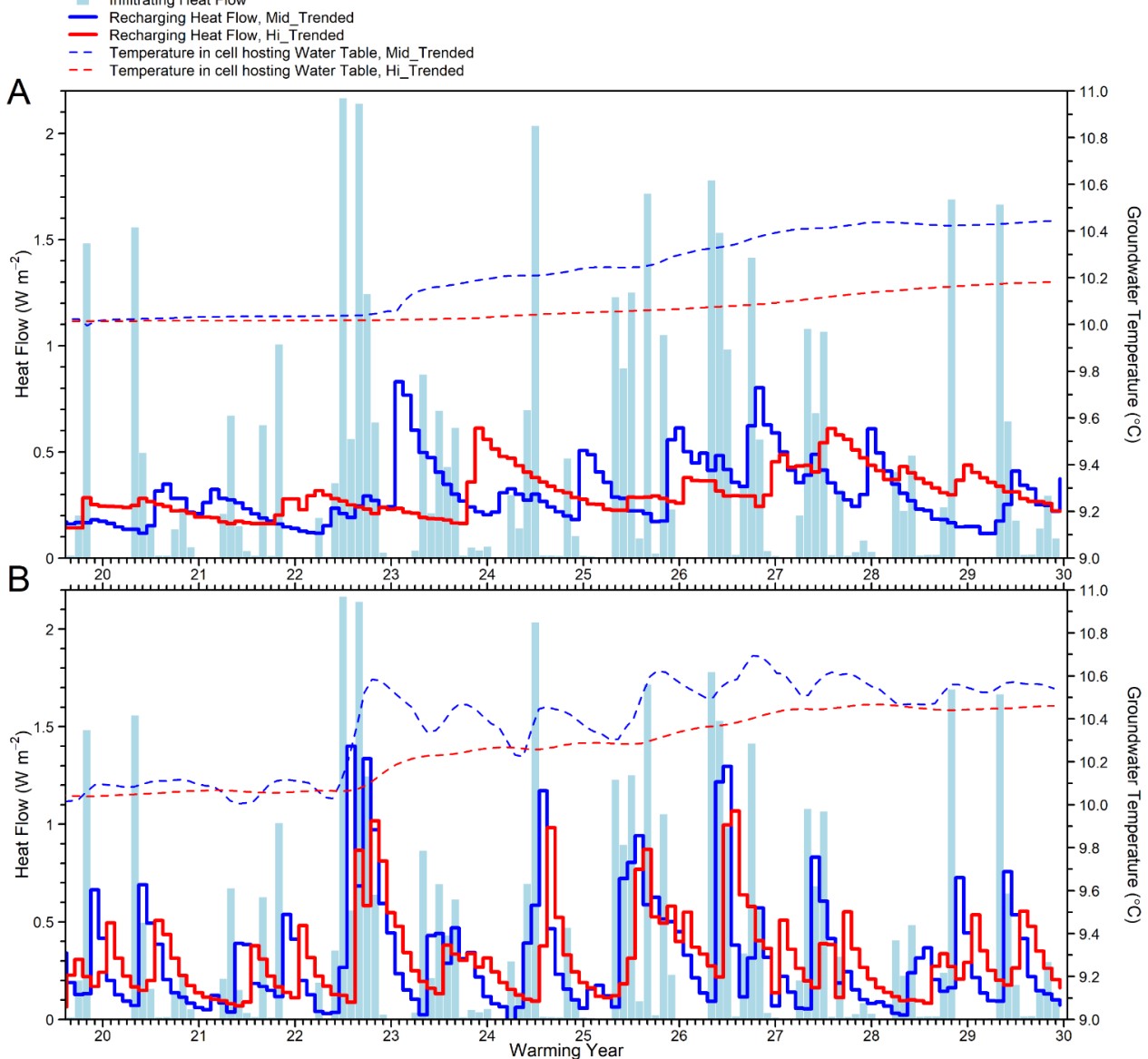

**Figure 11.** Convective heat flow (W/m$^2$) and temperature (°C) of the recharge at the (**A**) Upland and (**B**) Valley locations for the MID_TRENDED and HI_TRENDED simulations for the last 10 years of the warming period.

At the Upland location, the heat-flow lag in the MID_TRENDED simulation shown in Figure 11 is quite long, about 7–8 months. (See also Supplementary Material Table S3-2). In addition, the temperature response of the shallow groundwater in both the MID_

and HI_TRENDED simulations is noticeably subdued (Figure 11A), attributable to UZ thicknesses exceeding 40 ft (12 m; Figure 6A) and 100 ft (30 m; Figure 6B), respectively.

A clearer correspondence between the infiltrating and recharging heat flows is seen in the shallow groundwater at the Valley location where the UZ is relatively thin (Figure 11B). For example, the MID_TRENDED simulation, featuring relatively thin UZ thickness, shows strong heat-flow correlation to infiltration changes, peaking at a one-month lag, while the HI_TRENDED simulation, with relatively thick UZ thickness, shows a slightly less strong correlation peaking at a two-month lag (Supplementary Material Table S3-2). The water-table temperature time series at the Valley location, especially for the MID_TRENDED simulation, also shows a definite, if lagged, response to the infiltrating heat signal (Figure 11B). This responsiveness is due to reduced UZ thickness at this location, and, therefore, to reduced capacity for heat storage.

The local lagging and dampening of heat flow through the UZ evident in Figure 11 at the scale of a single water-table cell can be integrated over basins within the watershed by dividing the total heat flux recharging the basin by its area. In Figure 12A the heat-flow behavior over time in recharge is compared to the infiltration forcing at a small headwater basin upstream from Gage 235, 2 mile$^2$ (5 km$^2$) in extent (see Figure 3 and Table 1). The offset and attenuation of the infiltration forcing in the recharge time series over the last 10 years of warming, tends to be greater for the simulation with relatively thick UZ (HI_TRENDED) than the simulation with relatively thin UZ (MID_TRENDED), but both model versions show pronounced inertial effects due to the UZ. The lagging and dampening of heat flow in recharge corresponding to the entire stream and lake network on the east side of the domain (that is, to the area upstream of Gage 864, equal to 134 mile$^2$ (348 km$^2$)) is similar to that registered for the small headwater basin, although slightly more attenuated (i.e., Figure 12A versus Figure 12B). This similarity is a reflection of the spatially homogeneous conditions that obtain in the synthetic model.

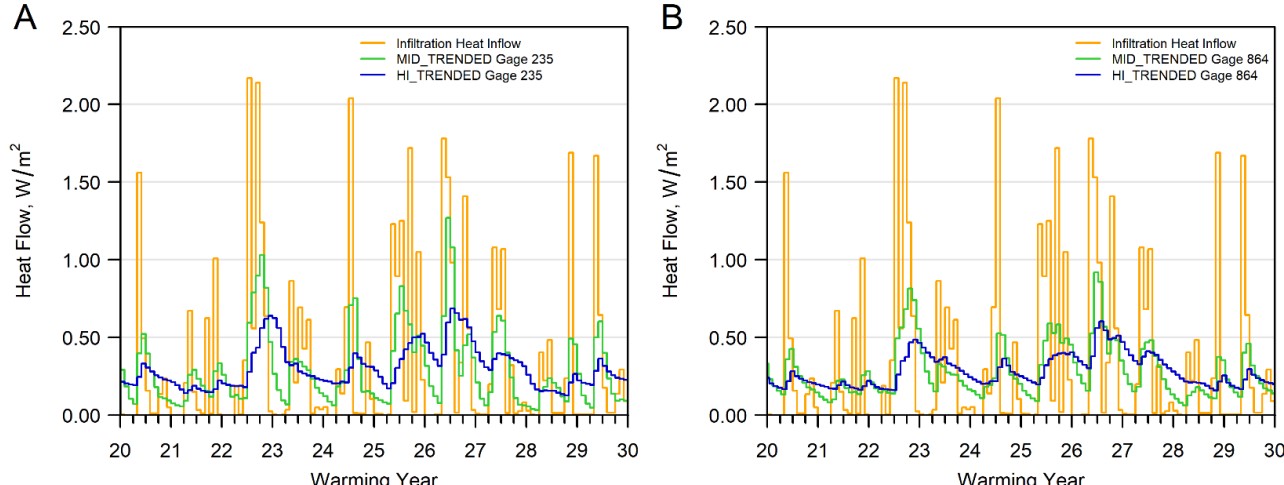

**Figure 12.** Heat flow response (Watts/m$^2$) along recharge pathway for (**A**) MID_TRENDED and (**B**) HI_TRENDED simulations. Graphs compare impulse heat flow in INFILTRATION to response heat flow in the pathway during last 10 years of warming for contributing basins corresponding to headwater gage (235) and model outlet gage (864). Heat Flow = Heat Flux over basin normalized by basin area.

Graphs similar to Figure 12 show the lagging and dampening behavior at the 235 and 864 gages for downgradient pathways associated with the stream interface (see Supplementary Information Figure S3-11b–f). For all pathways, the amount of lagging between the infiltration forcing and the downgradient heat flow or downgradient temperature response is quantified over a series of nested basins upstream of gage locations, according to the correlation method discussed in Supplementary Material Section S3. The tables

accompanying the graphs contain an array of these calculations. The amount of dampening for the same nested basins is also tabulated. Dampening is quantified based on a direct comparison between the average heat flow or temperature amplitude over the last 10 years of warming and the average amplitude over the last year of spin-up before warming. These pathway results constitute the key hypothetical findings arising from the deployment of the synthetic model.

Whereas distinct lag correlations between the infiltrating and recharging heat flows are evident along the UZ pathway, there is less coherence when considering the longer groundwater pathways that that terminates as direct groundwater discharge to streams (Supplementary Material Table S3-3). This weakening is attributed to mixing of shallow and deep groundwater flow paths and, to an expected lesser extent, changes in heat stored in the aquifer matrix.

The total streamflow carries the heat contribution of both the baseflow components and the storm runoff component. The temperature and heat flow in total streamflow at different gage locations along the stream network show little lag with respect to the infiltration temperature and heat flow ("From Inflow to Total Streamflow" row in Supplementary Material Table S3-3). This result is expected given that storm runoff is the dominant contributor of heat to the stream; that is, storm runoff carries heat quickly overland to the streams with minimal lag and dampening along its path (Supplementary Material Figure S3-10).

Watershed-scale dampening is expressed when incoming heat flow of the infiltration is partially stored in the UZ first, with additional heat storage occurring later in the groundwater system (Supplementary Material Section S3). Table 2 (top) and Figure 13 summarize the pathway dampening that occurs between the last year of spin-up and the last 10 years of warming in terms of heat flow at the basin outlet Gage 864. The climate forcing imposes a 31.3% average increase in the infiltrating heat flow within the contributing area of the basin over the warming period. The recharge transmits a fraction of that signal, producing about a 13% increase in heat flow relative to the spin-up conditions for both the MID_TRENDED and HI_TRENDED simulations. That is, the recharge delivers roughly four tenths of the warm-up entering at the top of the UZ to the water table. For down-system pathways, there is a further and rather sharp reduction in heat propagation through increased dampening of the original infiltrating heat signal. For example, where groundwater discharges to the stream, only about one tenth of the original infiltrating warm-up signal is simulated. There is comparatively less dampening when considering the total baseflow discharge; only about two tenths of the infiltrating warm-up signal is transported to the surface-water system. The baseflow to the stream network carries more of the climate forcing than the direct discharge component because it also includes heat flow from groundwater runoff, which is not subject to dampening. However, total streamflow at the outlet gage, because it incorporates undampened storm runoff, shows less dampening altogether–similar to that shown by recharge (about four tenths of the heat impulse–top of Table 2 and Figure 13). Storm runoff is simulated to be a powerful driver of stream heat-flow conditions.

**Table 2.** Dampening of warming along watershed pathways: Attenuation of convective heat flow and temperature along pathways contributing to the 864 gage location for MID_TRENDED and HI_TRENDED simulations.

| Heat Flow Pathway | Unit | Location | Model Version | Amplitude Percent Increase Due to Warming [1] |
|---|---|---|---|---|
| Infiltration | Heat flow (Watts/m$^2$) | Uniform across watershed | MID_Trended and HI_Trended | 31.3% |
| Recharge to Water Table | Heat flow (Watts/m$^2$) | Contributing Basin for Model Outlet (864) | MID_Trended HI_Trended | 13.3% 12.7% |
| Direct Discharge to Streams | Heat flow (Watts/m$^2$) | Contributing Basin for Model Outlet (864) | MID_Trended HI_Trended | 1.6% 3.7% |
| Baseflow to Streams | Heat flow (Watts/m$^2$) | Contributing Basin for Model Outlet (864) | MID_Trended HI_Trended | 6.8% 4.4% |
| Total Streamflow | Heat flow (Watts/m$^2$) | Contributing Basin for Model Outlet (864) | MID_Trended HI_Trended | 12.7% 11.1% |

| Heat Flow Pathway | Unit | Location | Model Version | Amplitude Percent Increase Due to Warming [1] |
|---|---|---|---|---|
| Infiltration | Flux-weighted temperature (°C) | Uniform across watershed | MID_Trended and HI_Trended | 22.7% |
| Direct Discharge to Streams | Flux-weighted temperature (°C) | Contributing Basin for Model Outlet (864) | MID_Trended HI_Trended | 1.3% 1.3% |
| Total Streamflow | Flux-weighted temperature (°C) | Contributing Basin for Model Outlet (864) | MID_Trended HI_Trended | 5.6% 5.4% |

Note: [1] Calculated from ratio of average values of last 10 years of warming to last one year of spinup.

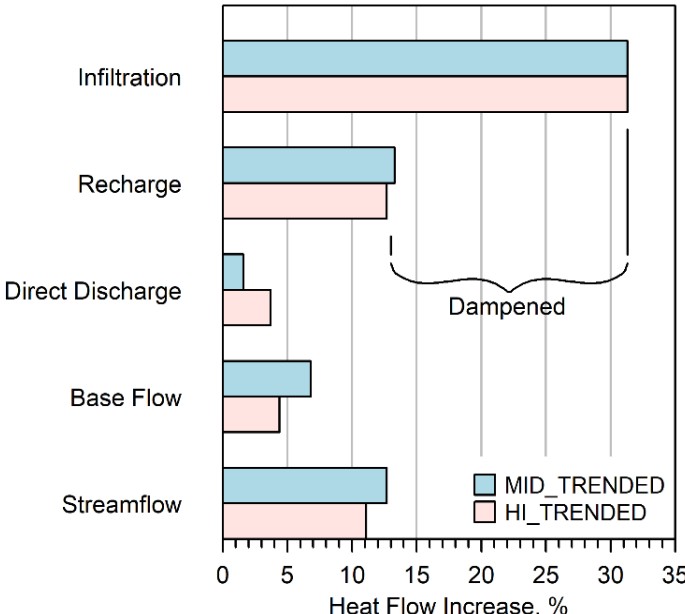

**Figure 13.** Changes to convective heat flow for different watershed compartments within the simulation are compared for the MID_TRENDED and HI_TRENDED model versions. An increase in heat flow is calculated as the average heat flow over the last 10 years of the of the warming period minus the average heat flow for last year of the spin-up period.

Dampening is also evident in the temperature response to the infiltrating heat flow impulse for down-system pathways (bottom of Table 2). Recall that the flux-weighted

temperature in the last year of spin-up averages 9.97 °C and rises to an average of 12.23 °C for the last 10 years of warming, equivalent to a 23% increase (taking 0 °C as the reference point). For both simulations, the direct groundwater discharge temperature increases by only 1.3% of the spin-up average condition; the corresponding relative increase in total streamflow temperature is about 5.5%.

*3.4. Implications for Modeling Watershed Heat Transport*

Several findings are important for watershed-scale heat transport simulations. First, changes to temporal dynamics of the system in the form of heat storage, lags, and dampening, can all be considered aspects of thermal inertia along watershed pathways. The synthetic model shows how the inertial strength of the UZ and the full groundwater system acts on phase responses to heat fluctuations and on amplitude responses to heat trends originating at the top of the system. The mitigating effect is opposed by the quick heat flows associated with groundwater runoff and, especially, storm runoff. The distinct inertial strengths of watershed pathways combine to produce the complex down-system baseflow and total streamflow responses.

Second, watershed heat transport must consider all heat transport pathways together to accurately simulate the complexities of the down-system response to warming infiltration. Consider the difference in system baseflow thermal response to warming as compared to total streamflow. The total streamflow thermal response is dominated by the amount of heat added to it by storm runoff, which during warm, wet months is conveyed rapidly (in the model context, instantaneously) to stream segments. A baseflow-only characterization would show much less effect of warming. Put otherwise: in the synthetic model, storm runoff constitutes only one-quarter of total streamflow in these simulations, but its thermal effect is disproportionally large due to the absence of any lag or dampening effects on its contributed heat load. Thus, the thermal load contributed by storm runoff overwhelms cooler thermal flows from direct groundwater discharge to the streams. It is worth noting, however, that there are periods of the year (often ecologically important) when storm runoff is largely absent and baseflow dominates total streamflow (and by extension its thermal regime).

Third, this modeling exercise was limited to evaluating the response to a high-emission climate scenario over 30 years. If the warming were to extend over a longer period, there is an expectation that the ability of the UZ and groundwater system to store heat would diminish over time and provide less dampening of the infiltrating heat flow before it reaches a down-system discharge location. In addition, watersheds with lower storage capacity, higher thermal conduction, lower thermal sorption, and higher UZ vertical hydraulic conductivity are expected to produce less lagging and dampening. Thus, the transferability of the results presented here should focus on the heat transport relationships between watershed components (e.g., the UZ, or the saturated zone) rather than on the specific percentages or absolute relative differences resulting from the use of the synthetic model.

*3.5. Limitations and Suggestions for Future Work*

The thrust of this study is to demonstrate that groundwater/surface-water models can be combined with climate scenarios to simulate water and heat flow at the watershed scale in ways that facilitate science-based forecasting of global warming effects on resources such as stream habitats. There are several limitations and lessons from our hypothetical study that may apply to future applications:

- The climate scenarios, appropriately downscaled, are a promising basis for forecasting effects of climate change on watersheds. For the synthetic model, we imposed a linear temperature rise in line with the high-emissions scenario for an area in the Upper Midwest, USA. No effort was made here to partition the expected heat inflow increase among the seasons or months–but this kind of refinement over and above simple linear infiltration trends might be warranted in a real-world application based on different statistical moments of the GCM results for the region under study.

- Similarly, the present study imposed a linear trend on future precipitation and infiltration: expected changes in precipitation intensity and seasonal distribution patterns were neglected but may be important for applications to real-world watersheds.
- A key expansion of the method presented in the companion paper [2] is the inclusion of storm runoff as part of the watershed flow and heat budgets. However, certain thermal mechanisms are still omitted, such as the effect of heat-bearing precipitation and solar radiation on streams as well as the latent heat effects due to evaporation from surface water bodies. Future developments could add these processes to the MT3D-USGS code if sufficiently important for calibration and forecasting.
- This study presents a lagging and dampening analysis of heat flows performed strictly in terms of the convective component. In a real-world application, this approximation, justified for the synthetic model, might not always prove adequate because of the particular importance of conductive and/or dispersive components at watershed interfaces. In such cases, it might be necessary to expand the heat-flow analysis to include all heat transport components, including possibly conduction and dispersion across streambeds. However, it is worth noting that the lagging and dampening analysis in terms of simulated temperature is not an approximation but reflects all heat transport components.
- In this study, the temperature of infiltration at the bottom of the root zone is set equal to the time series of the atmospheric temperature. The assumption may have its validity reduced with time steps shorter than a month or for seasons subject to high rates of evapotranspiration. Additional studies may be needed to determine if and at what time scale temperatures at the top of the UZ can be reliably equated with atmospheric conditions.
- The specific findings presented here regarding lags and dampening correspond to assumed uniform sandy subsurface conditions. In a heterogeneous setting with finer deposits and preferential flow, the phase and amplitude patterns might appreciably change (consider, for example, the effect of confining beds in the unsaturated and/or saturated systems).
- For the synthetic model under study, it was not necessary to impose a calibration period between spin-up and warming periods: only two periods, in this case both set to 30 years, were sufficient for demonstration purposes. However, a real-world application would likely include calibration to historical observations of heads, flows and temperatures. The length of the calibration period would depend on the available data but would need to be long enough to represent properly the transition from the dynamic equilibrium of the pre-calibration spin-up period to the more variable forcing during the calibration and prediction phases.
- If the model setup were modified to extend the warming trends incorporated in the heat inflow forcing function beyond 30 years, the amplitude of the energy and temperature effects over time would of course be magnified. Any applications of the method to real-world settings would likely simulate forecasts into the second half of the 21st century. Given the large degree of uncertainty around future thermal forcing, it is reasonable for applications of the proposed method to real-world watersheds that a range of GCM emission scenarios be considered (as is done in Hunt et al. [18,19]) to treat simulated heat flow findings in a more statistical fashion.
- Monthly time steps may not be sufficiently refined for some forecasts arising from climate warming (for example, fish vulnerability to short-term stream temperature fluctuations). In such cases, simulations with time discretization finer than a month might be warranted, though practitioners may consider restricting temporal refinement of the model to only those stress periods where it is needed.
- The surface-water network in the synthetic model is baseflow-dominated. Losing streams might be more common in a given real-world watershed, but both the MODFLOW and MT3D-USGS codes can handle any combination of gaining and losing conditions with respect to both water and heat flow.

- There are other simplifications adopted in this hypothetical approach that might require more attention in a real-world application. We assumed a no-flow boundary between an unconfined aquifer and underlying bedrock–in real-world settings, the possibility of water and heat loss below an unconfined aquifer might require explicit modeling of deeper units. We equated groundwater contributing area to stretches of stream completely within their topographic basins (Supplementary Material Figure S2-6)–in a real-world study the researcher might want to use the model to delineate groundwater divides more precisely based on the simulated flow system. We also assumed a small dispersivity value of about 0.91 m (3 ft) relative to a grid spacing of 91 m (300 ft). Higher values of dispersivity or identification of preferential pathways could have a strong influence on convection processes. Finally, an important limitation arises from the primitive lake physics in current versions of MODFLOW and MT3D-USGS, where mechanisms such as lake stratification, ice formation, and latent heat transfers during evaporation from surface water are neglected. Such lake processes continue to be subjects of active research that could lead to more sophisticated treatment of water bodies within the watershed thermal regime.

## 4. Conclusions

The objectives of this research were as follows:

(a) to forge a robust approach for applying numerical models to study the hydrologic effects of long-term climate change at the full watershed scale and at a monthly time interval, as deemed appropriate for taking account of how a warming trend imposed on background seasonal and random variability propagates through space from the top of the unsaturated zone downward;

(b) to use a synthetic model of a temperate watershed to not only develop the method but also to draw tentative conclusions about the degree of lagging and damping that a future climate forcing would undergo along distinct surface and subsurface pathways, resulting in predictable changes to the warming signal at unsaturated/saturated and groundwater/surface-water interfaces.

The first phase of this work demonstrated the utility of recent model enhancements for simulating how a climate signal is modified as water moves through the UZ and the groundwater system, as well as over the land surface, on its way to a surface-water network. The synthetic model was used to demonstrate the power of the widely used MODFLOW and MT3D-USGS software to track the watershed response to warming. The method yielded quantitative results for the transient distribution of heat flow conditions in the water table, as determined by the propagation of convective and conductive energy components, where it was shown that convection is more important than conduction for the simulated system. The method also allowed us to perform detailed impulse-response analyses of the convective heat signal integrated over time and its transient effect on the groundwater/surface-water system. The dominant effect of UZ thickness, highlighted in Morway et al. (2022b) [2], was confirmed when two model versions with different water-table depths at the watershed scale were applied to the study of heat-flow lags and dampening. The potential importance of the riparian zone was also evident when comparing the direct groundwater discharge response to the more integrated total baseflow response.

The time delays identified by the modeling exercise represented thermal inertia processes resulting from travel through the UZ and the presence of long flow pathlines in groundwater, as opposed to the quick flow resulting from groundwater discharge in riparian areas and storm runoff components. Lags in integrated response time for convective heat flows and for temperature of the streamflow were very short due to the large heat load carried rapidly to streams in warm wet months by undampened storm runoff. The imposed increase in the heat impulse at the top of the UZ was appreciably dampened along unsaturated, saturated, and surface-water pathways, but in complex ways. When the average convective heat flow in the last 10 years of a 30-year warming period was

compared to the dynamic equilibrium conditions at the onset of warming, the heat inflow signal was reduced at the water table to 40% the original signal. The presence of both short and long groundwater flow paths and variable path depths further reduced the strength of the thermal loading such that at the stream interface, it was a small fraction of the warming signal. However, when other components of the total baseflow to streams were considered (stormflow, rejected infiltration and groundwater discharge to riparian areas), the model simulated more efficient heat propagation, and the reductions in the warming trend relative to the initial impulse were similar to what was seen at the water table. The simulated dampening response in the streamflow itself could be evaluated in terms of both convective heat flow (diminished by roughly half at the watershed scale with respect to the initial warming impulse) and temperature (registering about one quarter the strength of the assumed near-surface rise).

Because not all parts of a watershed are equal from an ecological standpoint, future modeling studies will need to be tuned to simulate the lag and dampening effects of the subsurface system at interfaces of biological importance. For example, the thermal dynamics at the groundwater/surface-water interface will be of particular importance for a portion of the life cycle of some benthic invertebrates. A holistic watershed representation, i.e., one that includes the UZ, will likely prove useful for capturing complex water and heat flow interactions along the various watershed pathways and through interfaces of special importance.

**Supplementary Materials:** The following supporting information can be downloaded at: https://www.mdpi.com/article/10.3390/w14182810/s1.

**Author Contributions:** D.T.F., R.J.H. and E.D.M. shared equally in the conceptualization of the material. D.T.F. did most of the writing with help from R.J.H., while E.D.M. and Feinstein collaborated on the figures. D.T.F. took the lead on the response to reviewers with contributions from coauthors. All authors have read and agreed to the published version of the manuscript.

**Funding:** The authors would like to acknowledge support from the U.S. Geological Survey Land Change Science/Climate Research & Development for this work.

**Institutional Review Board Statement:** Not applicable.

**Informed Consent Statement:** Not applicable.

**Data Availability Statement:** The model executables, input and output files are available in an online model archive [33].

**Acknowledgments:** The authors acknowledge the assistance of Alden Provost (U.S. Geological Survey) and Vivek Bedekar (S.S. Papadopulos) in helping us to formulate the modeling approach. We remain especially grateful for the counsel and friendship of our recently departed colleague Richard Healy-he will be greatly missed by everyone who knew him. Any use of trade, firm, or product names is for descriptive purposes only and does not imply endorsement by the U.S. Government.

**Conflicts of Interest:** The authors have no conflict of interest.

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
