# Peer review of "Simulation of Heat Flow in a Synthetic Watershed: Lags and Dampening across Multiple Pathways under a Climate-Forcing Scenario"

_water, doi:10.3390/w14182810_

Round 1

Reviewer 1 Report

I have reviewed the draft paper: ‘Simulation of Heat Flow in a Synthetic Watershed: Lags and Dampening Across Multiple Pathways Under a Climate-Forcing Scenario’ submitted to a special issue of the journal Water. Fundamentally, the topic of heat transfer from the land surface to groundwater via advection/conduction, and then to adjacent surface waters via groundwater discharge and surface runoff under changing climactic conditions is a super important and understudied topic. The present study seems to represent a worthwhile incremental advance, using a simplified ‘sandbox’ watershed scale numerical model to explore high level concepts using new US Geological Survey modeling techniques. It seems the value in the study lies primarily in the demonstration of the modeling tools rather than the heat exchange process-based inferences, but the simulations do hint at important watershed heat storage, inertia, and transport processes that warrant much more focused study (perhaps with finer time step versions of the current model with subsurface heterogeneity). Beyond monthly timesteps and a homogenous subsurface, many important watershed heat fluxes do seem to be neglected here, such as heat fluxes associated with radiation exchange at land surface, sensible heat exchanges, and evaporative heat uptake from the land surface. Also, overland flow and interflow seem to be conflated in the paper, but they are indeed distinct processes. Interflow is typically defined as flow through the unsaturated zone, often lateral, which can discharge directly to the riparian zone and surface water bodies, and interflow seems neglected here or somehow combined with more simple overland flow. That is probably OK given the scope of the study, but overall the text is unclear or underexplained in many places regarding various heat transport processes and the specifics of the simulations. I have tried to point out many places that could use clarification or a bit more information in the attached commented version of the manuscript draft. For example, on first read I was unclear as to what period signals were being discussed in the context of heat transport timing lags through the unsat zone down to the water table. I assumed diel and event-based heat pulses/signals would be efficiently filtered with depth in the model so you would be focused on annual temperature signal transport. However, I think your are not focused on signal processing per se, but instead in the timing of warm months from land surface to the water table.  Without consulting the supplemental material in detail, I believe you are running the model at monthly timesteps, so the pules/lags being discussed are also at that coarse increment, but that is not clear. An opportunity may be being missed here to actually extract annual temperature signals at water table, groundwater discharge, and stream locations to perform a quantitative analysis of signal lag and damping, but perhaps that is out of scope for the current work that is centered around a numerical methods demonstration. Some of my comments/questions left early in the document are answered by information later in the document, but I left some of those review comments in place as the reader needs some additional detail up front in the introduction and abstract to understand the scope and goals of the study. I did not review the supplemental material in great detail. Overall, the paper figures are fairly coarse looking (beyond the expected low res review versions) and need a review for typos and other issues, some of which I have pointed out. Thank you for the opportunity for an early look at this interesting work.

‘Major’ comments:

-L341: The text states: ‘The flux of heat for any part of the model domain is calculated as the flux of water through the model cells constituting the given volume multiplied by the temperature of the water and by the density and heat capacity of fresh water’. Maybe I am missing something, but this would seem to only directly capture the advective heat flux and neglect conduction, yet the latter is an ongoing process wherever there is a temperature gradient. Can you explain this apparent discrepancy?

-The abstract could use some work to be a more clear summary of your work, and I have left several comments on the PDF document to that effect. Primarily, it would be good to give the reader a sense for the timescales on which you focus. Events, seasons, years, decades? What is the timestep of the model runs (monthly?)?

-I think it would be quite helpful to extract the groundwater discharge temperature over time at a few choice model cells and plot that relative to the land surface temperature pattern and downstream stream temperature, either at annual timescales or otherwise.

-This conceptual model seems to assume that all groundwater within a watershed discharges to surface water systems, but that is not really ever the case. What about heat transfer to deeper groundwater that does not discharge to the surface? Also, this study applies mainly to strongly gaining headwater stream systems, so you could make that more clear in the Intro and perhaps the title.

-The conceptual watershed heat transport model would seem to neglect quickflow/interflow.... which in my experience greatly exceeds actual land surface runoff. Also, what about heat loss via surface evaporation, is that flux accounted for somehow? The description of the model early in the text seems to neglect upward conduction of heat from groundwater through the unsat zone to the land surface when that surface is colder than groundwater, but that important process in mentioned in a later section, so I think that vertical heat loss to land surface during cold times is being tracked but could be better explained.

-In methods overland flow is mentioned as being synonymous with ‘interflow’. Its my understanding that Hortonian flow is across the land surface when rainfall exceeds infiltration capacity, while interflow is lateral through the unsat zone.... this terminology should definitely be clarified.

-On L135 this statement is made: ‘The amount of heat that enters the top of the UZ is the product of the infiltration rate and its temperature.’ This would oddly seem to neglect conductive heat exchange, which is influenced by advective heat exchange across the land surface interface, but also the temperature of that interface as influenced by direct radiational warming and cooling, evaporation, sensible heat exchange, etc. Can you rephrase or better explain your statement? Also, Does ET change over time, and is evaporation accounted for in the land surface heat budget?

-The end of the introduction may be a good place to clearly state your current research goals, rather than focusing on how the model capabilities differ from previously published models, as this work seems to be primarily pitched as process-based research rather than a model software update.

-Section 3.3 could use a nice figure to show simulated lags from a range of conditions (depth to water, etc).

-I am not sure if you are aware, but similar annual temperature signal timing lags between shallow aquifer and land surface have recently been used to infer relative shallow (from aquifer source depths that show pronounced annual temp signals) and deep (little to no annual temp signals) groundwater contributions to streams by comparing stream annual temperature signal characteristics to that of local air (e.g. doi.org/10.1038/s41467-021-21651-0, doi.org/10.1016/j.scitotenv.2018.04.344, https://www.sciencedirect.com/science/article/pii/S0075951117300592). Streams with apparent strong I am not suggesting that you need to cite these specific papers, but the concept could provide some nice context for your work from a baseflow and habitat resiliency perspective. Also, your current model could potentially be repurposed to quantitatively explore how shallow groundwater signals are translated to streams via groundwater discharge at the watershed scale… Have you extracted shallow groundwater, groundwater discharge, and stream annual temperature signals and compared their various phase and amplitude parameters?

-I do not think you need to directly account for the potential for enhanced ET in a warming climate, but you may want to reference research that predicts greater uptake of shallow groundwater by trees in some portions of the country under a warmer climate (doi.org/10.1038/s41467-020-14688-0), which in turn would influence heat transport to groundwater via percolation/conduction. Out of curiosity, what happens to heat in unsat zone water that is taken up by tree roots?

-Some of the figures looks a little unrefined with excel-based plots and instances such as ‘deg C’ for the units in an axis title instead of using the degree symbol. Figure 4 seems to be missing a legend for the various model layer colors. Figure 12 has a large label of ‘Dampenin’ with the lower portion of the ‘p’ cutoff, which I can only assume is a simple typo, but exemplifies the coarseness of many of the current figures.

-Please add a bullet in Methods to explain your sediment/rock thermal parameters. And it may be worth noting that a consistent porosity is highly conceptual at the watershed scale, and that parameter will impact heat transport in both the sat and unsat zone. Is there a bedrock layer in the model or do you only simulate unconsolidated sediments?

Author Response

Please see attachment (PDF file) for responses to inline comments.  It appears that only one file can be uploaded; I am inserting responses to general comments here:

Responses to Reviewer #1’s general comments for Water-180699

Simulation of Heat Flow in a Synthetic Watershed: Lags and Dampening Across Multiple Pathways Under a Climate-Forcing Scenario

Daniel T. Feinstein, Randall J. Hunt and Eric D. Morway

The general comments of Reviewer #1 were insightful and provoked important changes to the article. Our responses begin with the most significant modifications.

The reviewer pointed out that our analyses of lagging and dampening of the climate heat signal were restricted to convective component of heat flow and that the heat signal itself is quantified strictly as a convective flow, neglecting conductive and dispersive components. Both these statements are true: we did not clearly explain these aspects of our methodology in the original draft. We have added a fairly long discussion in section 3.2 of the manuscript that aims to overcome these omissions. The discussion explains and justifies the focus on convective heat flow at three interfaces – the top of the unsaturated zone (UZ), the water-table surface, and across streambeds.

The reviewer raised concerns that one potentially important pathway for heat flow was neglected in our analysis – namely interflow.  The term interflow has been widely interpreted since its first use in the 1940s (see Beven, 1989), and is typically generalized as subsurface contributions to streamflow that occur during a storm hydrograph (i.e., those that influence the hydrograph’s recessional limb).  Here, however, the reviewer focuses specifically on contributions to streamflow from lateral flow in the UZ which are one potential interflow mechanism. This terminology difference notwithstanding, we have revised the manuscript’s Introduction section to formally identify the potential for an interflow pathway and discuss why processes operating on the storm-hydrograph timescale are not explicitly included in our analysis of monthly to decadal heat transport on the watershed scale.

An important simplification incorporated in the proposed methodology is to equate the heat signal with infiltration that has already passed the root zone. Root zone processes in humid areas, which bear on both the movement of water and heat, include evaporation, transpiration, and conduction, leading to flow that can be both downward and upward. The key assumption in our methodology is that at a monthly transport time step, these root zone processes can be neglected and the warming signal at the top of the UZ can be equated with the average amount of water passing the root zone over the month and with the average monthly atmospheric temperature.  This assumption is discussed in detail in an article to be published concurrently in the Journal of Groundwater that deals with the mathematical and coding background to the method. However, a version of this statement has been added to the current manuscript in Section 2.2.

We have rewritten the Abstract and added language to the Introduction and Conclusions, all in an attempt to make our objectives clearer and to emphasize both the methodological and substantive parts of the study – the latter based on our generic findings about the degree of lagging and dampening exercises by different pathways through the watershed.

The reviewer also pointed our attention to recent publications that have documented lags in groundwater heat transport to streams, evidently at annual time scales. We have included these publications in the literature review part of our Introduction. We have also tried to sharpen our Objectives and Conclusion statement to make clear that this article has a somewhat different focus - to provide a method for investigating lagging and dampening in the presence of climate change simulated over several decades with emphasis on how different pathways in the subsurface, including the UZ, are expected to delay and attenuate the long-term climate signal.

Per the reviewers comment about the possibility of deep heat transport to bedrock, text has been added in the Abstract and Section 2.2 to state that the synthetic model under study consists of an unconfined sandy aquifer overlying effectively impermeable bedrock. In the Limitations section, language has been added to recognize that in a real-world application the model might have to take account of heat loss below the unconfined system.

The reviewer asked for discussion of the model flow and transport parameters in the Methods section of the manuscript (not just the Supplementary Material). Because there is targeted discussion of the parameters in a companion paper proposed for publication simultaneously with this article, we hope it is acceptable to refer the reader more forcefully to the companion article in this regard (text has been added).

The reviewer pointed out rightly that the surface-network is baseflow dominated. A statement to this effect has been added to Section 2.2 and a statement added to Section 3.5 (Limitations and Future Work) reiterating this might not be true in a real-world watershed, but that MODFLOW and MT3D-USGS can handle losing streams with respect to both water and heat flow.

Naturally, in reading a manuscript such as ours, which features a synthetic model and a generic analysis, it is proper to think about how it could be applied to real-world settings and what difficulties might emerge. We have tried to anticipate these concerns in the Limitations and Future Work section, listing mechanisms (such as evaporation from stream surfaces) and subsurface conditions (such as heterogeneity) which might complicate the analysis. However, it is our contention that most if not all these difficulties can be accommodated with enhancements to the proposed methodology.

The reviewer asked for more standardized graphs to accompany the manuscript text. The graphs have been upgraded and made uniform in appearance.

The reviewer wrote “-I think it would be quite helpful to extract the groundwater discharge temperature over time at a few choice model cells and plot that relative to the land surface temperature pattern and downstream stream temperature, either at annual timescales or otherwise.”  Figure 11 already showed the temperature response at two water-table cells (Upland and Valley) – but text has been added to call out these results more explicitly.

The reviewer remarked that Section 3.3 could use a “nice figure to show simulated lags from a range of conditions (depth to water, etc.).”  However, Fig. 11 in Section 3.3 shows exactly that. It is designed to contrast conditions at the Upland location (thick UZ) with those at the Valley location (thin UZ). The explanation points repeatedly to the effect of water-table depth on both the heat-flow and the temperature response in water-table recharge to the infiltrating heat signal. We now also include a plot (new Fig. 12) showing recharge lags and dampening that integrates the responses over areas rather than locations – specifically areas upstream of  two gage locations, one associated with a headwater, the other with a relatively large basin. The full set of these gage plots for all pathways is in the Supplementary Material.

Most of the reviewer’s in-line edits have been incorporated. In a few cases, justification is given for the original wording. The file “peer-review-20510460.v1.reviewer1_RESPONSES.pdf” contains the point-by-point responses.

We would like to express our gratitude for the breadth and depth of Reviewer1’s comments and hope that we have been responsive.

References cited:

Beven, K. (1989). Interflow. In: Morel-Seytoux, H.J. (eds) Unsaturated Flow in Hydrologic Modeling. NATO ASI Series, vol 275. Springer, Dordrecht. https://doi.org/10.1007/978-94-009-2352-2_7

Reviewer 2 Report

Reviewer Blind Comments to Author:

This research paper (water-1806991-peer-review-v1) deals with the simulation of heat flow modeling in a synthetic watershed. This paper presented systematic methodologies and physical factors for developing empirical models on the development of various pathways for groundwater discharge in variable climatic conditions.

The paper is one of the significant contributions to understanding the groundwater recharge flow in the natural watershed, which is most suitable for publication in Water. However, certain criticisms are observed in this article that needs further revision.

Abstract: There are significant corrections needed with respect to the technical writing of the Abstract.

Some issues need to be clarified as follows:

1.  What are the important factors that influence climatic conditions on the groundwater temperature at different levels below the earth's surface? This factor is essential in understanding the underground water flow system and should be discussed in detail.

2.  What is the size of the recharge area of the watershed in this study? Provide some statistical data to validate your numerical model.

3.  Research methodologies and their parameters are systematically organized.

4. Numerous grammatical corrections and re-organization of sentences are commented on in the main article in the form of annotation. That all points should be considered before submission of your revised version of the paper.

I feel this manuscript should be appropriate to recommend for publication in the Water with minor revision.

Reviewer 3 Report

Ms. Ref. No.: Micromachines-1757664

Title: Investigation of Multiphase flow in a Trifurcation Microchannel - A Benchmark Problem

In this paper, new capabilities for watershed-scale heat transport modeling are used to simulate thermal responses to climate warming across multiple pathways, including the unsaturated zone (UZ), in a synthetic watershed representative of temperate conditions. The simulations explore effects of variable unsaturated zone thickness, shallow/deep groundwater flow, and discharge to surface water on heat transport throughout the watershed. Lags simulated between infiltrating heat at the top of the UZ and the water-table response can exceed 6 months where the water table is deep (>10 m) but are on the order of 0-3 months when averaged over stream subbasins. However, only about 40% of the climate warming applied at the top of the UZ reached the water table due to heat storage in the UZ. The thermal loading in the subsurface is further dampened when discharged at the sediment-stream interface, where about 10% of the warm-up was seen. In contrast, the simulated warm-up in streamflow was similar to that observed at the water table. Stormflow runoff (an undampened pathway) was a significant contributor of heat to watershed streams despite contributing appreciably less water overall than baseflow.The analyses themselves are sound and I believe the results of their work. The data are well organized by the authors. I, therefore, recommend this paper be published in the Water Journal after the authors address the following comments.

·         Review English grammar as there are mistakes throughout the manuscript and the supplementary text. This article should be completely rewritten.

·         An abstract is not well organized. The abstract must be improved. The authors must explain the application and novelty of the research work add in the abstract section.

·         The literature section must be improved with more advanced articles and clearly why your present study is different, better to explain novelty.

·         More physical insight into the discussion section is needed.

·         The physical explanation of figures 10 -12 is limited. Please explain more

·         More physical insight into the discussion section is needed.

·      P:12, L_362: However, it is striking that for both gage locations in both the MID_ and HI_TRENDED 362 versions of the model), the heat fluxes entering the subsurface as infiltration, subsequently 363 converted to recharge, increases appreciably over the 30 years. Why?

In conclusion, this paper might be made suitable for publication in this Journal if the as-mentioned comments are clarified. These constitute a major revision of it.

Round 2

Reviewer 3 Report

The revise version could be accepted for publication.